

# Research on remote sensing image extraction based on deep learning

Zhao Shun*, Danyang Li*, Hongbo Jiang, Jiao Li, Ran Peng, Bin Lin, QinLi Liu, Xinyao Gong, Xingze Zheng and Tao Liu

Sichuan Agricultural University, College of Information Engineering, Yaan, Sichuan, China
* These authors contributed equally to this work.

## ABSTRACT

Remote sensing technology has the advantages of fast information acquisition, short cycle, and a wide detection range. It is frequently used in surface resource monitoring tasks. However, traditional remote sensing image segmentation technology cannot make full use of the rich spatial information of the image, the workload is too large, and the accuracy is not high enough. To address these problems, this study carried out atmospheric calibration, band combination, image fusion, and other data enhancement methods for Landsat 8 satellite remote sensing data to improve the data quality. In addition, deep learning is applied to remote-sensing image block segmentation. An asymmetric convolution-CBAM (AC-CBAM) module based on the convolutional block attention module is proposed. This optimization module of the integrated attention and sliding window prediction method is adopted to effectively improve the segmentation accuracy. In the experiment of test data, the mIoU, mAcc, and aAcc in this study reached 97.34%, 98.66%, and 98.67%, respectively, which is 1.44% higher than that of DNLNet (95.9%). The AC-CBAM module of this research provides a reference for deep learning to realize the automation of remote sensing land information extraction. The experimental code of our AC-CBAM module can be found at https://github.com/LinB203/remotesense.

# INTRODUCTION

Semantic segmentation is critical tasks when processing remote sensing images. It is used to segment objects on the earth, such as vegetation, roads, buildings, water bodies, and other objects. Its main purpose is to extract semantic information effectively and divide the studied image pixels into a number of object areas. Pixels belonging to the same category are classified as a specific homogeneous area while ensuring heterogeneity between different object areas. Traditional semantic segmentation methods are based on geometry and statistics, and before deep learning methods became popular, semantic segmentation methods based on traditional machine learning classifiers such as Texton Forest (*Shotton, Johnson & Cipolla, 2008*) and Random Forest (*Schroff, Criminisi & Zisserman, 2008*) were more commonly used. More recently, many models based on level sets (*Osher & Sethian, 1988*; *Ball & Bruce, 2007*; *Ma & Yang, 2009*), deep learning (*Längkvist et al., 2016*; *Zhao, Du & Emery, 2017*; *De et al., 2017*; *Tang et al., 2018*; *Zhao*

Corresponding author
Tao Liu, liutao@sicau.edu.cn

*et al., 2017*), and Markov random field models have been proposed (*Li, 1994*; *Besag, 1986*; *Nishii, 2003*; *Feng, Jia & Liu, 2010*; *Zheng, Zhang & Wang, 2017*).

In recent years, many techniques and prior information have been added to improve the accuracy of semantic segmentation technology. In particular, deep learning technology has greatly improved the performance of remote sensing image classification, remote sensing image retrieval, remote sensing image target detection, and other related technologies. Therefore, based on deep learning, the semantic segmentation of remote sensing images can be reasonably performed, the label of each pixel in the image can be predicted, and then be accurately classified. *Long, Shelhamer & Darrell (2015)* proposed a fully convolutional network (FCN) in 2014, applying deep learning to semantic segmentation for the first time, achieving end-to-end, pixel-to-pixel training, and combining traditional convolutional neural networks. The fully connected layer in the convolutional neural network (CNN) (*Kalchbrenner, Grefenstette & Blunsom, 2014*) is transformed into a convolutional layer, and the fully connected layer is removed to realize the input of any size image. The encoding part only obtains the feature layer through convolution and pooling processing, and the decoding part restores the feature map obtained by the last convolution layer through deconvolution and upsampling, thereby achieving pixel-level semantic segmentation. However, the use of FCN-based semantic segmentation has many problems such as chaotic sampling structure, loss of object details, or smoothness in the receptive field by fixed segmentation.

Therefore, in 2015, *Badrinarayanan, Kendall & Cipolla (2017)* proposed the SegNet algorithm. It uses a symmetrical structure to improve accuracy and efficiency, and uses pooling indices to save the contour information of the image, reducing the number of parameters. *Chen et al. (2014)* proposed the Deeplab algorithm and continuously optimized it on this basis. The v1 series not only considers the DCNN output but also considers the pixel value around the pixel when classifying each pixel so that the boundary of the semantic segmentation result is clear. To solve the problems caused by insufficient accuracy, pooling, and down-sampling repetition, the V2 series (*Chen et al., 2017*) chose to design a porous spatial pyramid pooling (ASPP) module based on the cavity convolution algorithm to expand the receptive field. V3 (*Chen et al., 2017*) further improved the ASPP module. There is also a v3+ series (*Chen et al., 2018*). One approach is to design a decode module based on v3 to refine the result, the other is to use modified Xception as the backbone. These are semantic segmentations based on the encoder-decoder (up-sampling/deconvolution) structure. *Cheng et al. (2021)* proposed a new training method, that was realized by adding a metric learning regularization term. *Cheng et al. (2018)* used context coding loss for canonical model training and proposed a novel feature enhancement network for detection. In addition, semantic segmentation models based on attention have been developed to obtain context information, such as non-local neural network (*Wang et al., 2018*) and DANet (*Xue et al., 2019*). These deep learning-based semantic segmentation algorithms can also be continuously optimized by ResNet (*He et al., 2016*) to continuously improve segmentation performance.

In this study, the method of introducing the attention mechanism into the semantic segmentation model is adopted to improve the accuracy. The attention mechanism was

first proposed in computer vision. In 2014, the Google Mind team used the attention mechanism (*Mnih, Heess & Graves, 2014*) on the RNN model for image classification. Since then, the attention mechanism has been widely used in computer vision fields such as target detection and semantic segmentation.

Subsequently, *Bahdanau, Cho & Bengio (2014)* used a mechanism similar to that used to simultaneously perform translation and alignment on machine translation tasks. Their work is regarded as the first to apply the attention mechanism to the field of NLP.

Attention mechanism is widely used in various NLP tasks based on neural network models such as RNN/CNN. In 2017, Google used the self-attention mechanism to learn text representations (*Vaswani et al., 2017*). Since then, the attention mechanism has become a research hotspot.

Aiming at the object classification and recognition of remote sensing images, this article addresses the problems of traditional remote sensing image extraction such as low accuracy, poor generalization ability, and low degree of automation. The original AC-CBAM module is created, and a full convolution based on the DNLNet encoding and decoding structure is constructed. A neural network is used for remote sensing image extraction, aiming to realize the precision and intelligence of remote sensing image extraction. Our main contributions are as follows:

1. Based on Landsat 8 OLI sensor data, this study preprocessed the image band. The preprocessed image is combined with the original image to form a 4-channel image as the input.

2. Based on the requirements of practical and experimental tasks, a new convolution attention module, AC-CBAM, is proposed in this paper. The innovative module combines asymmetric convolution and CBAM, which has better effect than CBAM and mainstream segmentation network in object classification and segmentation of remote sensing images.

3. Besides proposing the AC-CBAM module, this study analyzes how to effectively extract its features from the things, so that accuracy can be improved without a large number of search and experimental network structures. By analyzing the features of categories, in this study water and roads were found to have unique shape features, and, therefore, the idea of extracting features by using the uniqueness of asymmetric convolution is proposed.

4. We use sliding window prediction that ignores the edges to predict the entire remote sensing image, avoiding the problem of obvious splicing traces after the conventional grid cut prediction. Experiments show that the entire remote sensing image has a good prediction result.

## DATA PREPROCESSING

This chapter introduces the necessity and methods of remote sensing image preprocessing, gives the preprocessing results, and describes the difficulties and concerns of remote-sensing image block segmentation. Second, through Gaussian filtering, image fusion, and

band combination, the image quality and parcel information are enhanced, and the remote sensing image parcel dataset is prepared, which lays a good data foundation.

## Land parcel identification process

Remote sensing image classification technology provides a means to observe and analyze specific phenomena in the fields of commerce, ecology, forestry, and urban and regional planning. It has a wide application prospects in these fields. Land parcel segmentation using in-depth learning technology for remote sensing images is an important research topic in all fields. Precise land parcel extraction results will provide further research in all fields. This has great significance. In a real application scenario, the result of land parcel information extraction is highly required, but there are many interference factors in the land parcel information extraction of remote sensing images. For example, there is more cloud cover in the remote sensing image, and some areas have distinct brightness differences on the visual level, different bands perform differently on water, and in some bands, land and water have the same visual performance, as well as the shadows of land objects, which all increase the difficulty of land parcel extraction. If the remote sensing image is simply preprocessed, instead of enhancing and highlighting the land parcel information, a large number of pixel misclassifications will often occur. This causes unsatisfactory land extraction results and fails in practical application scenarios. In addition to the above interference factors, the remote sensing image itself has more complex and diverse feature information. Classifying water bodies, vegetation, buildings, roads and other objects from these complex feature information not only requires many human and material resources, but also makes it difficult to ensure extraction accuracy. For efficiency and accuracy of land parcel extraction, this study excludes the ground object information outside water, vegetation, buildings and roads. It only focuses on the land parcel information of remote sensing images, simplifies the task of remote sensing image recognition, focuses on the land parcel extraction of remote sensing images, and simplifies the multi-classification problem of remote sensing images into five classification problems of water, vegetation, buildings, roads and others while improving the efficiency and reduces the workload of human visual interpretation, and improves the efficiency and accuracy of remote sensing image block extraction.

The neural network obtains the global optimal weight matrix by forward error conduction and reverse weight updating. The trained matrix is used to predict the remote sensing image and obtain the final land parcel prediction result. Remote sensing land parcel identification is a pixel-level classification task, which classifies each pixel point into five categories: water, vegetation, buildings, roads, and others. The specific process of land parcel identification is as follows: (1) the remote sensing image is input into the trained network model; (2) the output of the model is the confidence level at which the pixel points belong to five categories, and the pixel point category is determined according to the confidence level; (3) the color is filled according to the decision category of the pixel points to achieve five classifications of the land masses of the remote sensing image; (4) the classification results are smoothed and denoised by the median filter to obtain the

**Table 1 Study the basic information table of the image.**

| The study area | Image acquisition time | Cloud content (%) |
|---|---|---|
| Kunming Dianchi Lake, Fuxian Lake, Yangzonghai | 2020.1.18 | 0.03 |
| Pan Yang Lake | 2019.1.23 | 1.6 |
| Leshan city, Yibin city area | 2018.4.18 | 0.09 |

final remote sensing land mass prediction results. The output of the two-dimensional median filter is:

$$g(x, y) = \text{med}\{f(x - k, y - l), (k, l \in W)\} \tag{1}$$

where $f(x, y)$ and $g(x, y)$ are the original image and the processed image respectively.

## Study area

We chose Kunming in southwestern China as our research city. The longitude and latitude of Kunming are 102°10′ ~ 103°40′E and 24°23′ ~ 26°22′N. The research area includes three large lakes: Dianchi Lake, Fuxian Lake and Yangzonghai Lake. In this study, the Minjiang River and part of the Jinsha River water bodies in Leshan City were selected. The longitude and latitude were 102°15′ ~ 104°15′E and 28°28 ~ 29°56′N. The mixed complex region mainly refers to the region with complex block combinations, with latitude and longitude ranging from 115°47′ to 116°45′E and 28°22′ to 29°45′N. Blocks in this region were used to test the comprehensive extraction ability of the model. Poyang Lake in southeast China was selected as the study area. The relevant information is presented in Table 1.

The data used in this study are from the Landsat 8 satellite data. Landsat 8 remote sensing satellite was launched by NASA in 2013. Compared with Landsat 7, Landsat 8 only adjusts the bands of the data, eliminating the effects of atmospheric scattering and absorption, water vapor and other factors on the data. In addition, two new bands, blue band (band 1 0.433–0.453 μm) and short-band infrared band (band 9 1.360–1.390 μm), have been added. The original image of remote sensing data is shown in Fig. 1.

Landsat 8 satellite data are widely used in geological exploration, environmental management, resource assessment and other domains, providing a large amount of remote sensing observational data for various disciplines and related research. It has a total of eleven bands, including eight multispectral bands, one panchromatic band and two thermal infrared bands, which can address different application scenarios. The main parameters are listed in Tables 2 and 3.

In this study, the red, green, and near-infrared bands were selected from Landsat 8 OLI data and fused with 15-meter panchromatic band to enhance resolution, covering three research areas. In this study, automatic extraction and manual verification were used to complete the plot labeling. The image label after manual interpretation was accurate enough, so we took the result of plot marking as the mapping label of the sample image and the baseline of the test image.

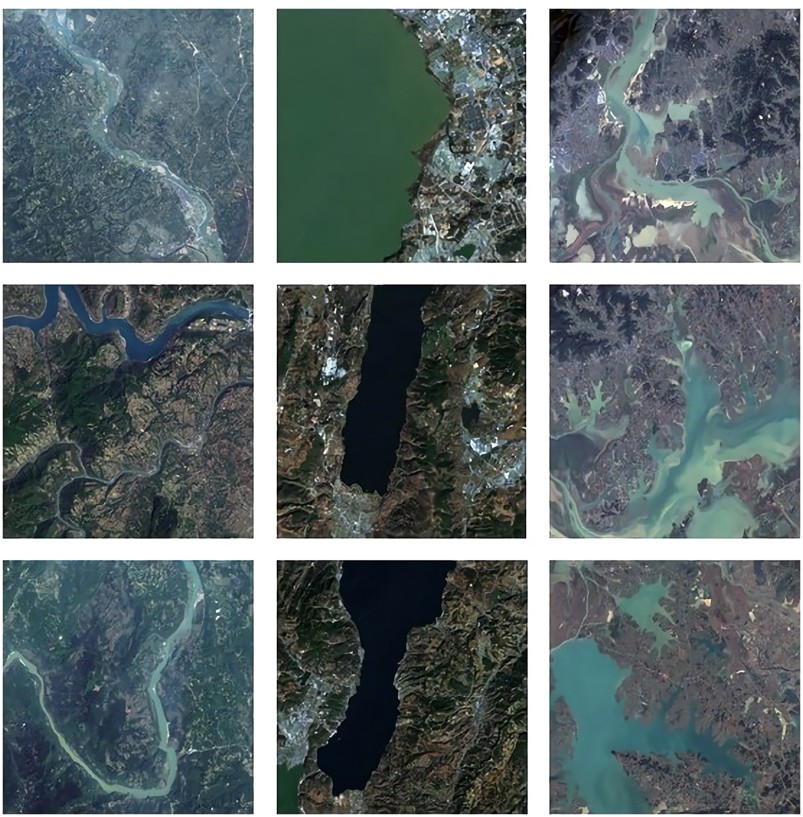

**Figure 1 The original image.**     

**Table 2 Landsat 8 satellite sensor parameters.**

| The sensor | Band | Wavelength range/μm | Signal to noise ratio | Spatial resolution/m |
|---|---|---|---|---|
| OLI | 1-COASTAL/AEROSOL | 0.43-0.45 | 130 | 30 |
| | 2-Blue | 0.45-0.51 | 130 | 30 |
| | 3-Green | 0.53-0.59 | 100 | 30 |
| | 4-Red | 0.64-0.67 | 90 | 30 |
| | 5-NIR | 0.85-0.88 | 90 | 30 |
| | 6-SWIR1 | 1.57-1.65 | 100 | 30 |
| | 7-SWIR2 | 2.11-2.29 | 100 | 30 |
| | 8-PAN | 0.50-0.68 | 80 | 15 |
| | 9-Cirrus | 1.36-1.38 | 50 | 30 |
| | 10-TIR | 10.60-11.19 | 0.4k | 100 |
| TIRS | 11-TIR | 11.50-12.51 | 0.4k | 100 |

## Image preprocessing

Remote sensing image preprocessing involves eliminating and suppressing some influences and errors in imaging, which mainly come from external conditions during imaging, such as the satellite sensor itself, absorption and scattering of ultraviolet radiation

**Table 3 Data product parameters.**

| The parameter types | Values |
| --- | --- |
| The product type | Level 1T terrain correction image |
| The output format | GeoTIFF |
| Sampling methods | Cubic convolution algorithm |
| Map projection | UTM-WGS84 projected coordinate system |
| Fastest return cycle | >72 h |
| Angle | 98.2 degrees |
| Operation cycle | 98.9 minutes |
| Rail type | Near polar solar synchronous orbit |
| Track height | 705km |

by atmosphere, solar altitude angle, earth surface topography and other factors. To obtain real reflectance data, methods such as radiometric calibration and atmospheric correction are used to preprocess remote sensing images to obtain real image data. In addition, in order to obtain richer remote sensing information and higher-quality images, image enhancement is first carried out on remote sensing images to highlight image edges and improve the overall quality of data products, thus laying a data foundation for subsequent data set division, thus improving the accuracy of plot discrimination.

A Gaussian filter is a linear smoothing filter that is widely used in image processing. It is mainly used to suppress noise and smooth images. Because noise is transmitted, it is a severe problem in the later application of digital images. Then most of the noise in the image is Gaussian noise, so the Gaussian filter is widely used in image denoising.

For remote sensing classification tasks, the selected size of Gaussian kernel is $7 \times 7$, the average pixel weight of the Gaussian kernel is larger, and the surrounding pixels are smaller, which reduces the importance of surrounding pixels and reduces image blur during anti-aliasing. The main function of Gaussian filtering is to smooth the image. Based on this, the original image with twice the size can be subtracted from the image after Gaussian filtering, and the resulting image has more obvious block edges.

Means of improving the image quality based on existing Landsat 8 data must be considered because the resolution of the satellite data is limited. In the data, the resolution of the panchromatic band is higher than that of multispectral data, so image fusion is used to fuse multispectral data to improve the image resolution. Brovey transformation fusion, which is a simple and effective method, is mainly used for the process. The processed image not only highlights the spatial resolution of the panchromatic band data, but also highlights the differences in the features of the objects in different bands.

While being generated, remote sensing images will be distorted to a certain extent owing to the influence of the satellite-sensor tilt angle, atmospheric scattering, water vapor, and other factors. To produce remote sensing images as realistically as possible, certain methods must be used to preprocess them to inhibit and eliminate interference. The image contents and different research objectives require different pretreatment methods.

### Radiometric calibration and atmospheric correction

Remote sensing technology receives electromagnetic wave information reflected by various objects on the Earth surface from outer space and carry out attribute analysis and imaging. The satellite sensor receives the solar radiation reflected from the Earth surface, and the final original image is a synthesis of all information due to atmospheric scattering, water vapor absorption, surrounding objects and other factors. The process of atmospheric correction is the process of separating specific spectral information from the study an object's attributes. Because there are errors in sensor imaging, these errors need to be eliminated when using remote sensing images for analysis and research, which is the process of radiation correction. The remote sensing data processing software used in this study is ENVI 5.3, through which remote sensing images can be atmospherically corrected. The ENVI FLAASH module was used for the atmospheric correction of remote sensing images after radiometric calibration.

### Remote sensing image enhancement

The main purpose of remote sensing image enhancement is to enhance the image quality and information, so that the plot information can be observed more clearly and intuitively under the observation of human eyes, which plays a crucial role in the discrimination of plots. Common remote sensing image enhancement methods include image synthesis, numerical filtering and contrast stretching. Different enhancement methods have been adopted to match diverse research content and methods. In this study, numerical filtering was used to enhance image quality, band combination was used to enhance image block information, and the effect of block discrimination is strengthened.

(1) Image enhancement in spatial domain. In this study, a Gaussian low-pass filter was used to denoise and smooth the image. A low-pass filter is mainly used to carry out two-dimensional convolution calculations through a Gaussian checking image.

$7 \times 7$ Gaussian kernel is adopted in this paper, and the specific structure is shown in Fig. 2.

Gaussian low-pass filtering processing flow:

a) Reading of various bands required.
b) The selected band is processed by the Gaussian filter.
c) The Gaussian kernel is moved across each pixel in the image, and the pixel value of the corresponding pixel is convolved by the kernel.
d) The calculated value is taken as the final pixel value.

The actual processing results are shown in Fig. 3.

(2) Band combination image enhancement. Generally, color images are RGB false color images. In remote sensing image classification, different RGB components are selected according to different classification targets to combine new standard false color images. Landsat8 has many different bands, and the combination of different bands also has different uses. The specific band combination scheme and uses are shown in Table 4.

| 0.0000 | 0.0000 | 0.0003 | 0.0006 | 0.0003 | 0.0000 | 0.0000 |
|--------|--------|--------|--------|--------|--------|--------|
| 0.0000 | 0.0011 | 0.0079 | 0.0153 | 0.0079 | 0.0011 | 0.0000 |
| 0.0000 | 0.0079 | 0.0563 | 0.1082 | 0.0563 | 0.0079 | 0.0003 |
| 0.0000 | 0.0153 | 0.1082 | 0.2079 | 0.1082 | 0.0153 | 0.0006 |
| 0.0000 | 0.0079 | 0.0563 | 0.1082 | 0.0563 | 0.0079 | 0.0003 |
| 0.0000 | 0.0011 | 0.0079 | 0.0153 | 0.0079 | 0.0011 | 0.0000 |
| 0.0000 | 0.0000 | 0.0003 | 0.0006 | 0.0003 | 0.0000 | 0.0000 |

**Figure 2** Gauss kernel structure.

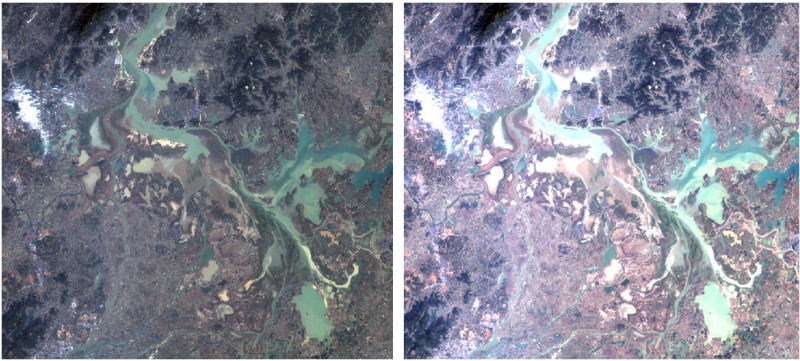

**Figure 3** Comparison before and after enhancement.

**Table 4 Band combination scheme table.**

| RGB combination | Main uses |
|-----------------|-----------|
| 4 (Red), 3 (Green), 2 (Blue) | Natural true color |
| 7 (SWIR2), 6 (SWIR1), 4 (Red) | City |
| 5 (NIR), 4 (Red), 3 (Green) | Standard false color images, vegetation, water |
| 6 (SWIR1), 4 (NIR), 3 (Blue) | Agricultural |
| 7 (SWIR2), 6 (SWIR1), 5 (NIR) | Penetrating the atmosphere, surveying special geological structures |
| 5 (NIR), 6 (SWIR1), 2 (Blue) | Healthy vegetation |
| 5 (NIR), 6 (SWIR1), 4 (Red) | Land |
| 7 (SWIR2), 5 (NIR), 3 (Green) | Remove the natural surface of the atmospheric image |
| 7 (SWIR2), 5 (NIR), 4 (Red) | Short wave infrared |
| 6 (SWIR1), 5 (NIR), 4 (Red) | Vegetation analysis |

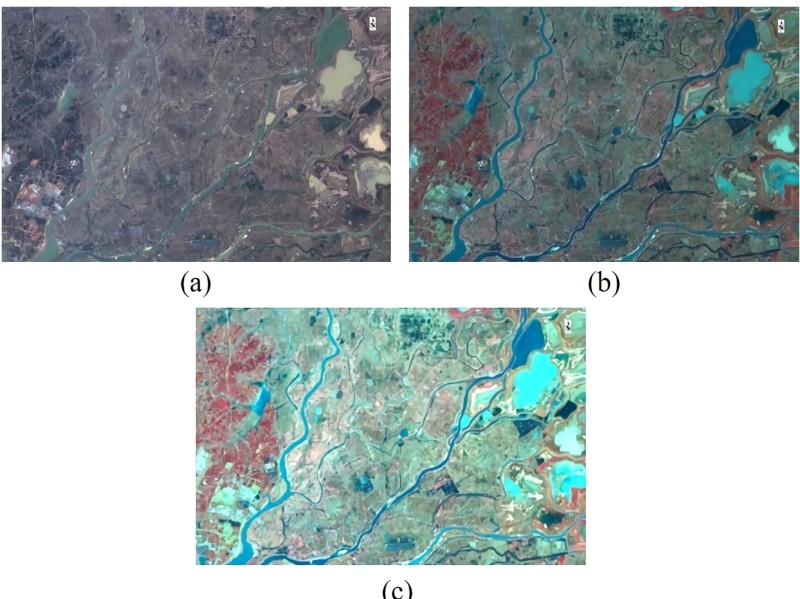

(a)  (b)

(c)

**Figure 4 Before and after band combination with Gaussian filtering.** (A) Original image, (B) the image after band combination, and (C) the image after Gaussian filtering.

For Landsat 8 OLI remote sensing products with atmospheric correction, which provide a variety of band data, including panchromatic, near-infrared, and green bands, we can easily produce standard false-color image data. The green, red, and infrared bands of the remote sensing data are assigned to the blue, green and red band in RGB respectively. It is well known that the colors of vegetation and water are similar in unprocessed remote sensing images. By synthesizing standard false color images, vegetation will be predominantly red, water will be green, blue, dark blue, and so on, depending on the number of microorganisms inside. One of the best benefits of converting remote sensing images to standard false color images is to distinguish vegetation from the color of water. According to the above description, we use five (NIR), four (Red) and three (Green) bands to enhance the band combination of remote sensing images and enhance the land parcel information. The results are shown in Fig. 4.

### Image fusion

Because the resolution of multispectral data is low, and the panchromatic band has a better resolution than multispectral data the panchromatic band was fused with the selected multispectral band. The new images have improved resolution, retain their multispectral characteristics, and the image quality and performance are enhanced.

There are information differences among the satellite sensors. To eliminate such differences, composite models are generally used to integrate data from different sensors to obtain clearer images and more pronounced ground object information. This is more conducive to remote sensing classification. In the case of limited existing data, it is of great practical significance to make full use of existing data to extract ground object information from remote sensing images. In a real scene, data from different sources fill the entire

experimental stage, and data from different sources have different advantages and disadvantages. Image fusion can effectively complement the advantages of data and compensate for the shortcomings of a single data source.

Landsat 8 satellite multi-spectral data resolution is 30 m, band 8 is panchromatic and the band resolution is 15 m, the selection of multi-spectral data band 5 (near infrared), band 4 (red), band 3 (green), and panchromatic band image fusion can effectively improve image clarity and data quality. Existing image fusion methods include HIS transform, multiplication transform fusion, wavelet transform fusion and Brovey transform fusion. The Brovey ratio transform fusion method was used in this study.

The Brovey ratio transformation and fusion method is a multiplicative band operation with a panchromatic band after the multispectral data has undergone normalized processing, and it clearly represents surface objects such as mountains, vegetation, and water bodies. This fusion method can improve the segmentation effect, and is, therefore, used in this study.

In this study, the Brovey transform fusion method uses the image and panchromatic bands of the red band (R), green band (G), and near-infrared (NIR) to conduct fusion, finally obtaining good spatial resolution from the information and retaining multispectral images. This method also promotes the extraction of detail.

The fusion formula of Brovey transformation is as follows:

$$\begin{cases} BT_{red} = Pan \times \dfrac{R}{R+G+NIR} \\[2mm] BT_{gre} = Pan \times \dfrac{R}{R+G+NIR} \\[2mm] BT_{nir} = Pan \times \dfrac{NIR}{R+G+NIR} \end{cases} \tag{2}$$

$BT$red, $BT$gre, and $BT$nir respectively are pixel values of red, green, and near-infrared images after fusion. Where, R, G, and NIR are pixel values of red, green, and near-infrared bands in multi-spectral images respectively, and Pan is a pixel value of panchromatic band of high-resolution images. Images are processed according to the Brovey formula above, which improves visibility to the human eye, sharpens plot features and achieve higher image quality.

In this study, only Landsat 8 satellite data were used as the data source, and Brovey transformation was used for image fusion of partial multispectral band data and panchromatic bands in the image. This effect is shown in Fig. 5.

### Dataset preprocessing

The convolutional neural network has certain standards for the size of the input image, and the size should be standard and uniform. Too large a size is not conducive to the extraction of detailed features by the convolutional layer, and too little feature information extracted from too small a size is not conducive to pixel classification. The particularity of the convolutional operation requires that the length and width of the input feature map be consistent. Therefore, it is critical to determine the input end of the model. If the size of

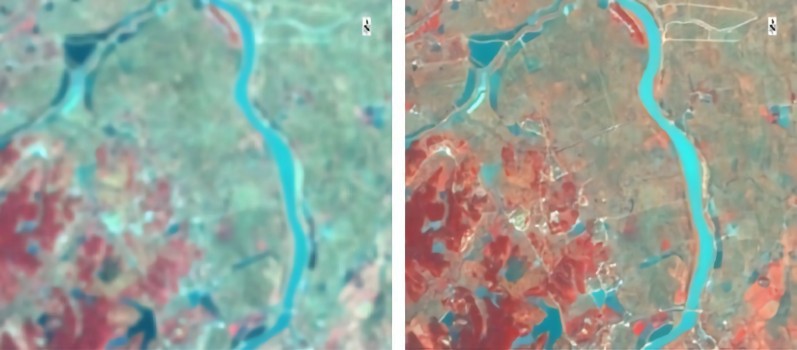

**Figure 5 Before and after fusion.** The left is the image before fusion, and the right is the image after fusion.               

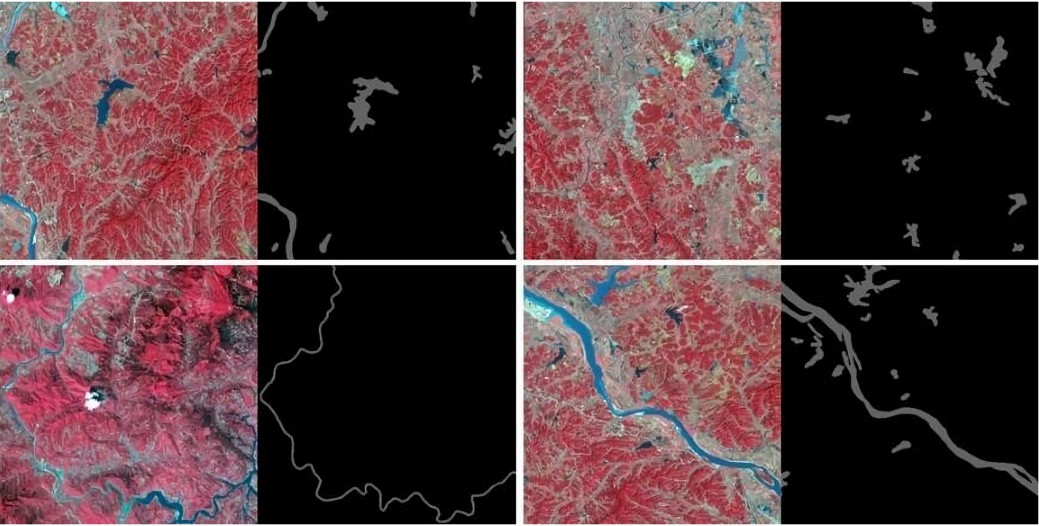

**Figure 6 Visualization of training set image and corresponding label taking water body as an example.**               

the input end is larger, some image details will be lost, and if the size of the input end is smaller, training errors will be generated because many blocks of a certain type, thus affecting the accuracy of the model. As the spatial resolution of the remote sensing image data obtained from Landsat 8 OLI satellite is 30 m, it has medium resolution. Considering the image input size and label distribution comprehensively, image cutting should be performed first. The cutting method randomly selects coordinates, which are taken as the origin of interception, and sampling interception is carried out. Large remote sensing images and corresponding labeled images are cut into sub-images in the above way, with sizes of 512 × 512, 256 × 256 and 128 × 128. Figure 6 shows only a sample image of the water segmentation.

Remote sensing images are too large to be placed directly into the neural network, and the number of remote sensing images is small. Therefore, OpenCV is used to expand and

add noise to the images, to effectively improve the training accuracy and generalization ability of the network. The main processing steps are as follows:

1. The original and labeled images were rotated by 90, 180, and 270 degrees along the Y-axis.
2. Noise, including salt and pepper noise, Gaussian noise, and other noise transformations were added to the original image transformation.
3. Light transformation was performed on the original image, including for brightness.
4. The original image was processed using fuzzy methods.

After image cutting and data expansion, the amount of data reached 10,000 pieces. In addition to image expansion, corresponding label images were expanded. After expansion, the classification distribution of the dataset was balanced, and the image resolution was sufficient to meet the demands of training the model. After the cutting image was trained, the prediction image was output. The predicted images were backtracked to the origin of the cutting and were individually spliced into the original large image. After a simple adjustment, the final prediction remote sensing image was obtained.

In this study, 512 pixels were selected as the reference scale, which, because of its rich background information, is the most appropriate scale for the human eye to recognize differences. We divide the complete and test images into three levels, namely 512 × 512, 256 × 256, 128 × 128, thus obtaining the initial sample set and test set. See Fig. 7.

We selected three scene images for data augmentation processing. Because the neural network is sensitive to data with different directions, different colors, and with or without noise points, we expand each image to a photo set with 128 × 128, 256 × 256, 512 × 512 sizes by randomly cutting, angle rotating and adding noise points. Then, for the 128 × 128 data graph, set the category proportion threshold to 50%, delete the images whose category proportion exceeds the threshold, and form the following training set, as shown in Table 5.

For Landsat 8 OLI sensor data, the first four bands of image preprocessing obtain real remote sensing images, by removing the interference of the satellite sensor itself as much as possible. The tagged data produced by the third national census data of water conservancy projects are automatically checked after expanding and dealing with the noise of the dataset, for it to have a certain size and diversity.

## METHOD

### Attention mechanism of computer vision

In human vision, the visual attention mechanism processes brain signals. Human vision functions by rapidly scanning the global image to focus on the target area. Therefore, this area should receive more attention to obtain detailed information about the target and suppress irrelevant information. For example, when people view pictures, they tend to focus on noteworthy objects but pay less attention to the background. There are three common types of attention mechanism models: the spatial attention model, channel attention model, and spatial and channel mixed attention model.

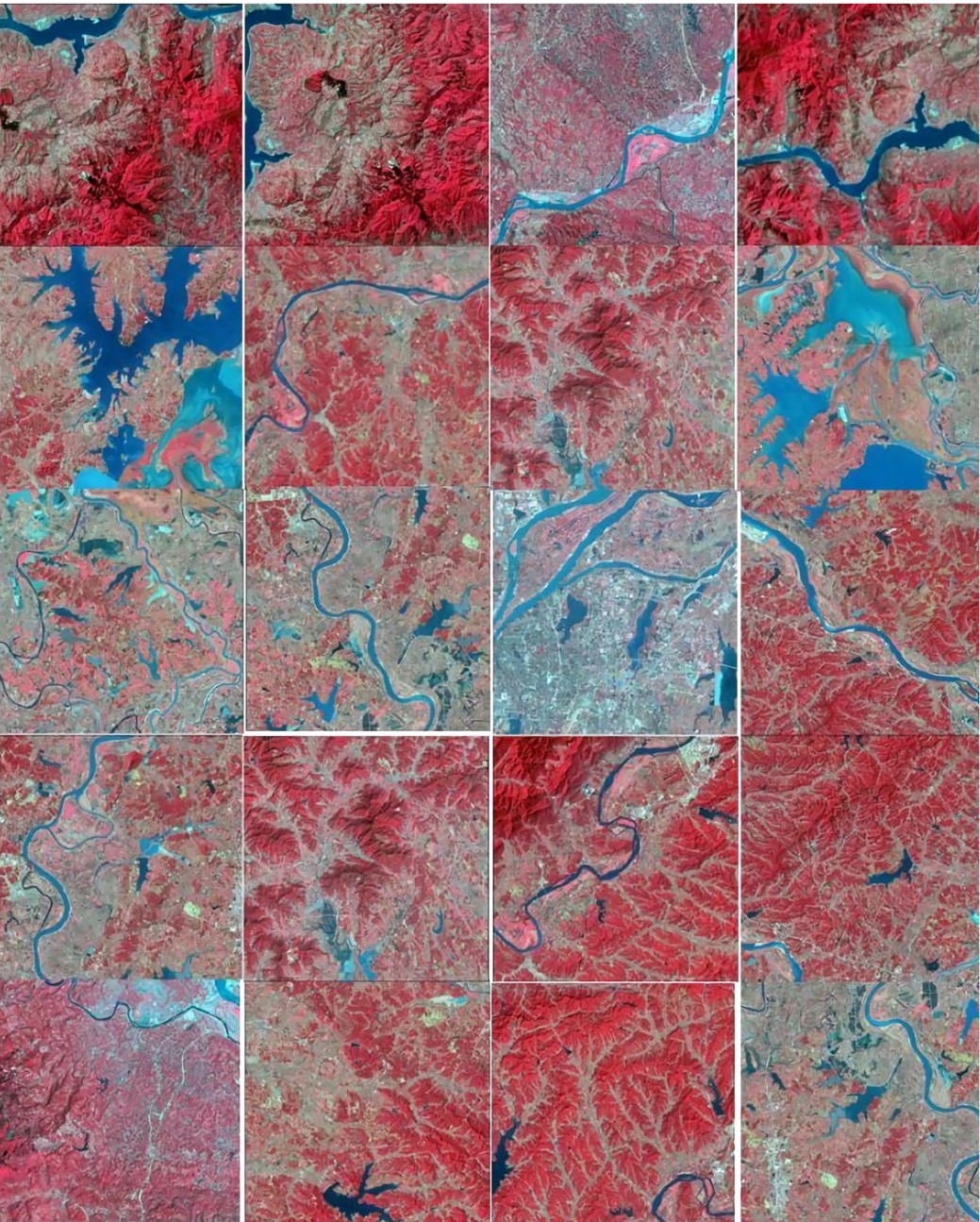

**Figure 7 Part of the training set image processed by data enhancement.**

**Table 5 Number of samples at different scales.**

| Image size | Number |
| --- | --- |
| 128 × 128 | 5000 |
| 256 × 256 | 3000 |
| 512 × 512 | 2000 |

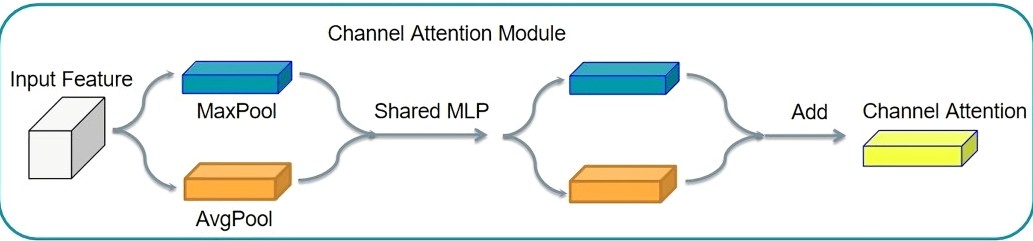

**Figure 8  Channel attention mechanism structure.**  

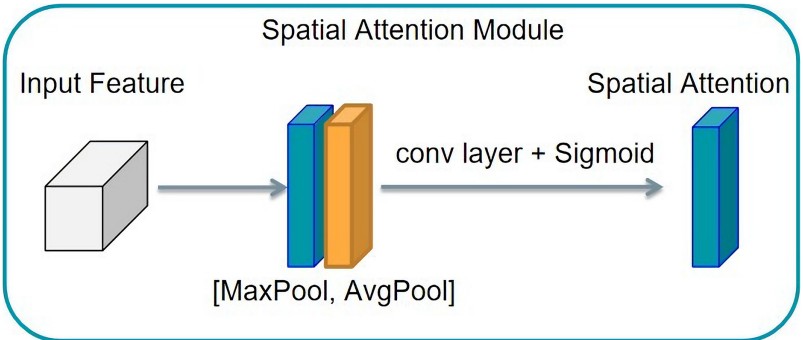

**Figure 9  Spatial attention mechanism structure.**  

## Channel attention

A good example of the channel attention mechanism is SENet (squeeze and congestion net). The SENet principle involves squeezing the tensor size. Then, it is changed into a tensor of $1 \times 1 \times C$, for the feature graph of each channel to become a real number, thus completing feature compression. The compressed tensor was used for the excitation operation. The excitation operation uses the parameter W to generate weights for each characteristic channel. The parameter W is learned to explicitly model the correlation between feature channels, thereby realizing the weight distribution for each channel. SENet uses two layers of bottlenecks to multiply the corresponding elements. Thus, the output tensor with the characteristic of the attention distribution in the channel can be obtained. This structure is shown in Fig. 8.

SENet can achieve notable network performance improvements with minimal additional computation. Because this idea can be applied to various existing network structures, SENet has many derivatives. A typical combination is ResNet and SENet.

## Spatial attention

The spatial attention mechanism differs slightly from the channel attention mechanism. If the channel attention mechanism achieves the distribution of attention along the channel, the spatial attention mechanism obtains the spatial distribution of attention, known as the distribution on a two-dimensional picture. This structure is shown in Fig. 9. Then a convolution operation is used to transform the feature graph with the number of channels two into the feature graph with the number of channels. To realize the principle of the spatial attention mechanism a feature graph with a channel number of two

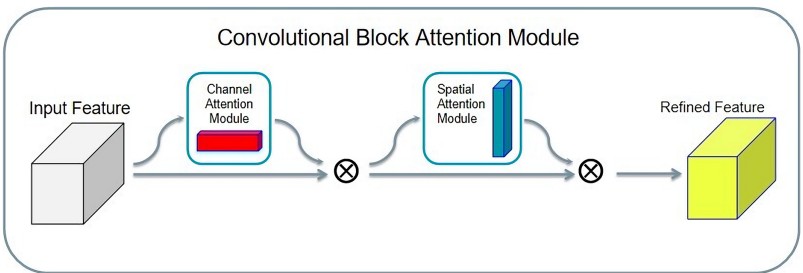

**Figure 10 Convolution block attention module structure.**

must first be obtained from the channel dimension by compression. Then, a convolution operation is used to transform the feature graph with two channels into the feature graph with one channel. This obtains the spatial distribution of attention. The eigenmatrix is then multiplied by the corresponding elements of the input tensor to obtain the output tensor characterized by the attention distribution along the channel.

### Integration of channel and spatial attention mechanism

The convolution block attention module (CBAM) is a simple and effective attention module. Given an input tensor, the CBAM takes it in two directions: spatial attention and channel attention. Finally, the attentional feature distribution is multiplied by the input tensor to achieve adaptive feature optimization. The structure of CBAM is shown in Fig. 10.

The advantage of the CBAM is that it is lightweight, and, considering its general purpose, it can be integrated into any convolutional neural network. Second, compared with SENet, CBAM adopts both global average pooling and global maximum pooling, and experiments confirm that, when simultaneously using two types of pooling, the CBAM has a better effect than only using one type of pooling.

## Asymmetric convolution

Asymmetric convolution operates with different lengths and widths of the convolution kernel. Christian Szegedy's Rethinking Inception Architecture for computer vision is an example of asymmetric convolution. Its structure is shown in Fig. 11. Compared to ordinary convolution, the significance of designing asymmetric convolution lies in several factors. First, the asymmetric convolution with the convolution kernel size of n × 1 and then the asymmetric convolution with a convolution kernel size of 1 × n is equivalent to the result of convolution with the convolution kernel size of n × n. Meanwhile, asymmetric convolution requires only 2 × n multiplications, while ordinary convolution requires n × n multiplications. Therefore, the larger convolution kernel requires less computation.

## AC-bottleneck

The basic bottleneck structure is shown in Fig. 12. First, it reduces the dimensions using a 1 × 1 convolution kernel. Then, it is convolved with a 3 × 3 convolution kernel. Finally, the

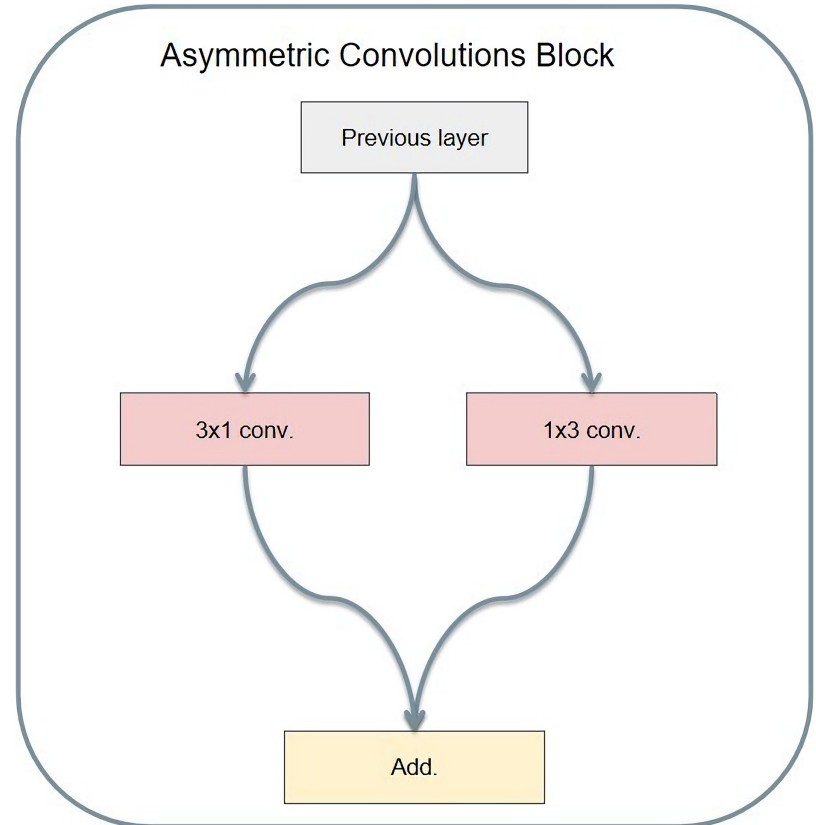

**Figure 11 Asymmetric convolution module structure.**

dimension is raised by a convolution kernel of size $1 \times 1$. The main function is to deepen the network and reduce the amount of calculation by reducing the number of parameters. Feature extraction is more effective after reducing the dimensionality.

After extensive experiments, the Swish activation function has proved superior to ReLU. However, compared with ReLU, it has the disadvantage of a higher computational overhead. To reduce the computational overhead of Swish, H-swish was proposed. Therefore, in this study, H-swish directly replaces the ReLU activation function.

$$h - swish(x) = x\frac{ReLU6(x+3)}{6} \tag{3}$$

The input tensors were separately convolved asymmetrically twice, and the two results were added. Then, the result activated by the h-swish function is passed to the CBAM. Finally, the result was added to the residuals to obtain the final result. Asymmetric convolution, the CBAM, and residual structure can greatly improve the training effect with a small amount of computational overhead, as shown in Fig. 13. Compared with the CBAM, the AC-CBAM introduced an h-swish function. The introduction of the h-swish function reduced the computational cost of asymmetric convolution and increased the computational speed. In addition, the AC-CBAM added two asymmetric convolutions in front of the CBAM module compared with the unmodified CBAM. In remote sensing

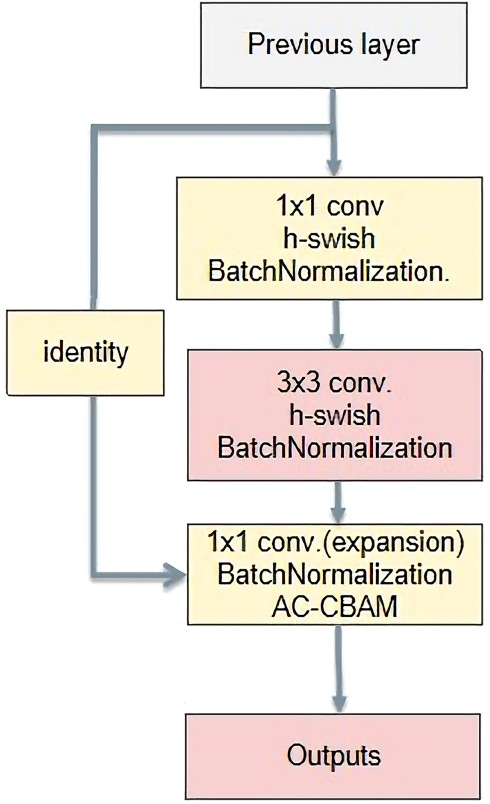

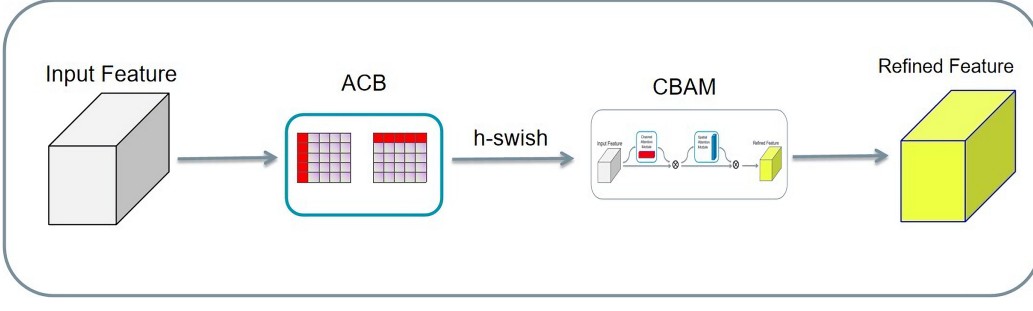

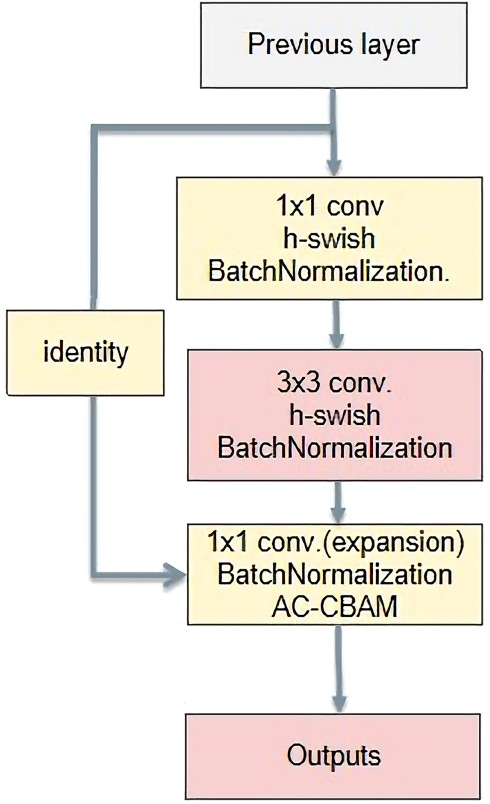

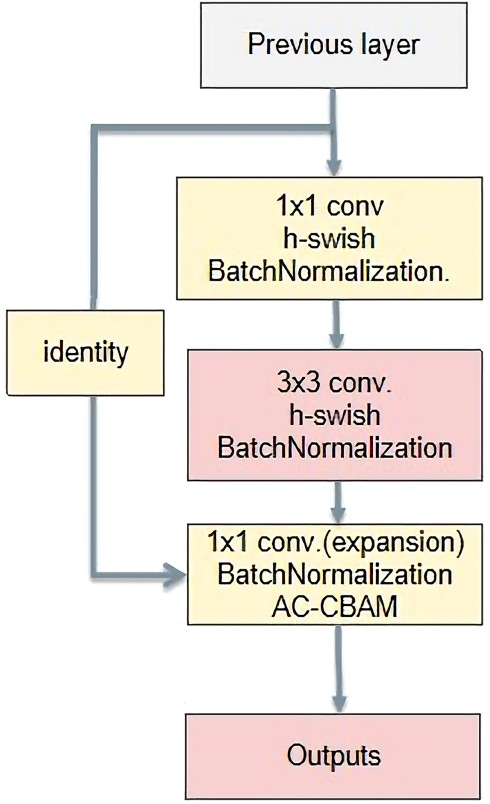

**Figure 12 A new bottleneck structure developed in this paper.**

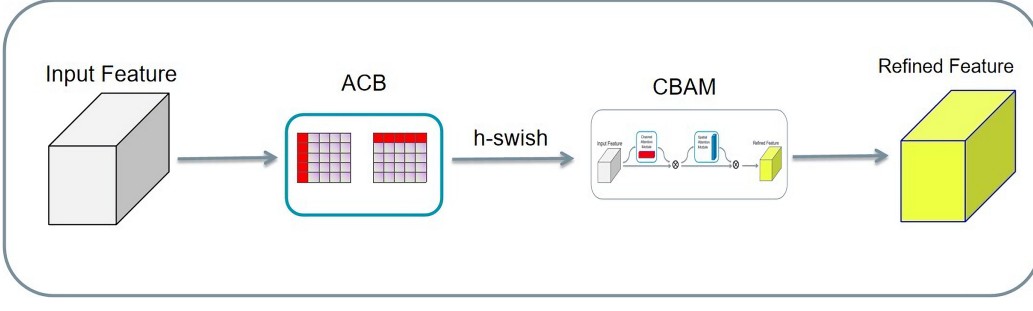

**Figure 13 A new AC-CBAM structure.**

images, water and roads were found to differ from others features, being mainly in long strip shapes. The other features are mainly square, requiring the long-strip image features to be extracted using asymmetric convolution.

# EXPERIMENTAL PROCESS

## Experimental settings

The program in this study adopts Pytorch, the mainstream deep learning framework, and the hardware and software configurations of the machine are shown in Table 6.

**Table 6 Hardware and software configuration table.**

| Configuration project | Detail |
| --- | --- |
| CPU | Intel (R) Xeon (R) Gold 5218 CPU @ 2.30 GHz |
| RAM | 128 G |
| Graphics card | 2 × NVIDIA Quadro RTX 5000 |
| Operating system | 64-bit Windows 10 |
| CUDA | Cuda 10.2 |
| Data processing | Python 3.8 |

**Table 7 Network model parameter list.**

| Parameters | Value |
| --- | --- |
| epochs | 150 |
| batch_size | 4 |
| activation | h-swish |
| Input_tensor | (512, 512, 4) |

The parameters of the deep learning network model used in this study are shown in Table 7.

The ratio of the training set to the validation set in the dataset was 4:1. The initial learning rate was set to 0.01. At the same time, due to the limitations of the experimental environment in this paper, the batch size and iteration times (epoch) were set to 4 and 300, respectively, and the performance was optimal. The optimizer selects SGD because it converges to the global optimal solution.

## Experimental results

Experiment 1: Three CBAM modules were added to the convolution layer. Simultaneously, the ReLU function is replaced by the h-swish function.

Experiment 2: After adding three CBAM modules to the convolution layer, the third CBAM module was replaced with the AC-CBAM module. Simultaneously the ReLU function is replaced by the h-swish function.

Experiment 3: The AC-CBAM module was added only after the third convolution layer. Simultaneously, the ReLU function is replaced by the h-swish function.

Experiment 4: Based on Experiment 3, the AC-CBAM of the last convolution was replaced by the CBAM.

The corresponding structure is shown in Fig. 14.

The relevant experimental results are shown in Tables 8 and 9.

By comparing Tables 8 and 9, it can be concluded from Experiments 1 and 2 is that under the same conditions, adding the CBAM module to the first two convolutions and the AC-CBAM module to the last convolution in Bottleneck is better than adding the CBAM module to the third convolution.

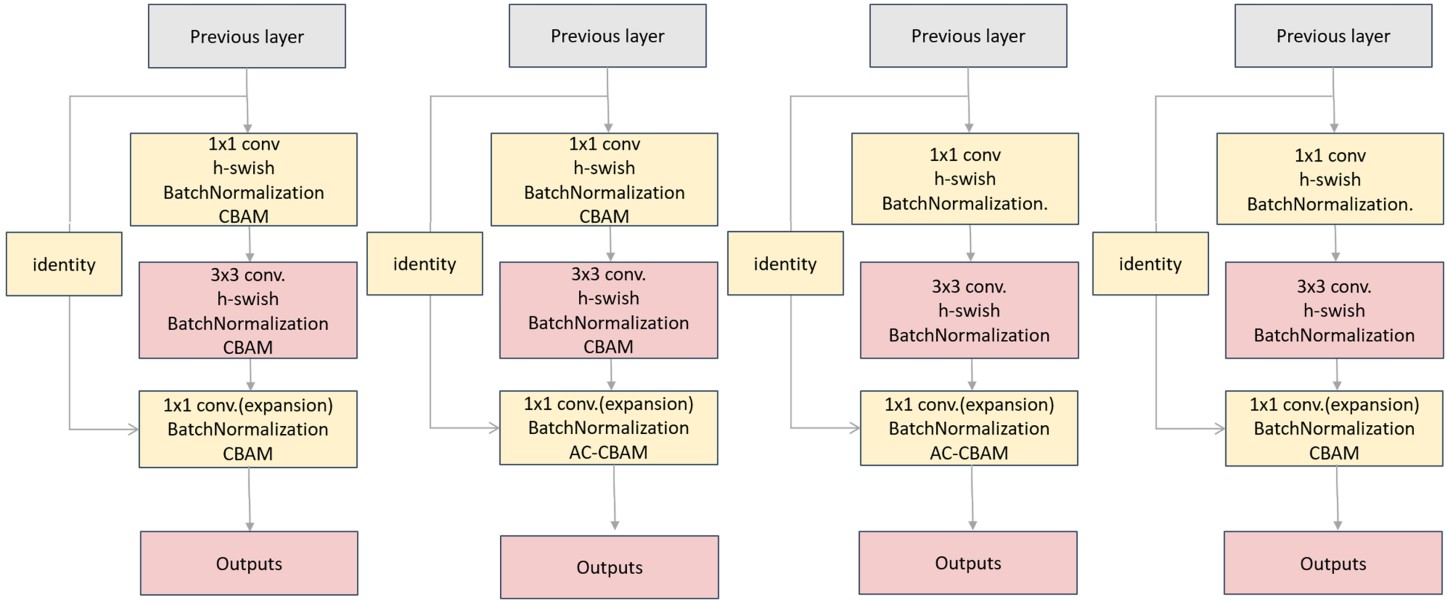

**Figure 14 Comparison of the three botttleneck diagrams of the experiments.** Experiment 1 is on the left, Experiment 2 in the middle and Experiment 3 on the right.

**Table 8 Results of three experimental schemes.**

| Experiment 1 | | | Experiment 2 | | |
|---|---|---|---|---|---|
| **Class** | **IoU** | **Acc** | **Class** | **IoU** | **Acc** |
| others | 95.65 | 97.93 | others | 94.35 | 96.97 |
| forest | 98.33 | 99.14 | forest | 97.8 | 98.79 |
| road | 97.34 | 98.47 | road | 96.52 | 98.24 |
| building | 98.01 | 98.91 | building | 97.27 | 98.74 |
| water | 95.43 | 97.84 | water | 93.88 | 97.31 |
| **Experiment 3** | | | **Experiment 4** | | |
| **Class** | **IoU** | **Acc** | **Class** | **IoU** | **Acc** |
| others | 96.21 | 97.97 | others | 76.21 | 84.41 |
| forest | 98.58 | 99.3 | forest | 87.63 | 93.67 |
| road | 97.68 | 98.87 | road | 83.33 | 92.29 |
| building | 98.32 | 99.19 | building | 88.79 | 94.69 |
| water | 95.9 | 97.99 | water | 79.08 | 89.49 |

**Note:**
IoU represents the intersection ratio between image prediction results and real labels, and Acc represents the accuracy of pixel prediction.

Conclusions can be drawn from Experiments 1 and 3, confirm that, under the same other conditions, the effect of not adding CBAM to the first two convolutions and adding AC-CBAM to the last convolution is better than that of adding CBAM to the first two

**Table 9 Average accuracy of the three schemes.**

| Experiment 1 | | | | Experiment 2 | | | |
|---|---|---|---|---|---|---|---|
| Scope | mIoU | mAcc | aAcc | Scope | mIoU | mAcc | aAcc |
| global | 96.95 | 98.46 | 98.46 | global | 95.96 | 98.01 | 97.97 |
| Experiment 3 | | | | Experiment 4 | | | |
| Scope | mIoU | mAcc | aAcc | Scope | mIoU | mAcc | aAcc |
| global | 97.34 | 98.66 | 98.67 | global | 83.01 | 90.91 | 90.35 |

Note:
mIoU represents the average IoU of all categories, mAcc represents the proportion of correctly classified categories calculated separately, and aAcc represents the average mAcc of each category.

**Table 10 Relevant models are compared with AC-CBAM accuracy.**

| Model | Backbone | mIOU | IoU.others | IoU.forest | IoU.road | IoU.building | IoU.water |
|---|---|---|---|---|---|---|---|
| PointRend (*Kirillov et al., 2020*) | ResNet101 | 0.948 | 0.926 | 0.972 | 0.953 | 0.956 | 0.924 |
| OCRNet (*Yuan, Chen & Wang, 2020*) | HRNet48 | 0.936 | 0.906 | 0.966 | 0.938 | 0.959 | 0.912 |
| GCNet (*Peng et al., 2017*) | ResNet101 | 0.940 | 0.914 | 0.968 | 0.944 | 0.961 | 0.916 |
| Deeplabv3+ (*Chen et al., 2018*) | ResNet101 | 0.945 | 0.921 | 0.975 | 0.949 | 0.963 | 0.922 |
| DANet (*Wu et al., 2021*) | ResNet101 | 0.948 | 0.926 | 0.972 | 0.953 | 0.965 | 0.924 |
| APCNet (*He et al., 2019*) | ResNet101 | 0.952 | 0.933 | 0.974 | 0.959 | 0.969 | 0.929 |
| ANN (*Zhu et al., 2019*) | ResNet101 | 0.934 | 0.903 | 0.965 | 0.936 | 0.959 | 0.907 |
| DNLNet (*Yin et al., 2020*) | ResNet101 | 0.959 | 0.942 | 0.978 | 0.964 | 0.973 | 0.937 |
| Ours AC-CBAM | ResNet101 | **0.973** | **0.962** | **0.986** | **0.977** | **0.983** | **0.959** |

Note:
Relevant models are compared with AC-CBAM accuracy, where IoU. A represents the value of IoU in category A.

convolutions and adding AC-CBAM to the last convolution. AC-CBAM alone was found to be more effective.

The conclusions drawn from Experiments 3 and 4, confirm that under the same conditions, the effect of adding the AC-CBAM to the bottleneck structure is better than that of the CBAM.

The conclusions drawn from Experiments 2 and 4, confirm that under the same conditions, the effect of adding the CBAM and AC-CBAM in the bottleneck is inferior to adding only the AC-CBAM. To test the robustness and effectiveness of the model in this study, we compared the SOTA model proposed in recent years with AC-CBAM, and the results are shown in Table 10. The mIoU of the AC-CBAM model reached 97.34%, which is 1.44% higher than that of DNLNet (95.9%), and the IoU of each category was the highest.

## Slide window prediction

In the process of prediction, the remote sensing image is too large to be directly predicted, so it is necessary to segment the remote sensing image and predict the segmented sub-

blocks. Finally, the prediction image is spliced according to the segmentation sequence to obtain the prediction image of the original remote sensing image.

However, through local amplification, we found that this effect was not ideal. Because of the problem of the receptive field of the convolutional network, attention was focused on the middle region, and the segmentation effect was poor for the edges of small blocks of images, and the sense of abruptness after fusion was obvious. Using only a simple segmentation prediction an image will exhibit a strong edge with the high error rate of edge prediction causing obvious abrupt changes, and other shortcomings. Therefore, the sliding window prediction method was used for enhancement.

In sliding window prediction, the overlapping sliding window strategy (sliding window step size < sliding window size) was adopted. In prediction, only the central region of the prediction results is retained, and the edges of images with inaccurate predictions are discarded. In this study, we make a slide-window prediction based on the segmentation of the original image and obtain a good prediction effect.

We set the length of the original image as x and the height as y, and the relevant filling and sliding window settings are as follows:

1. Fill 1 (yellow part): Fill the lower right boundary to an integer multiple of the size of the sliding window prediction window to facilitate divisible cutting;

$$\text{Padding\_x\_yellow} = \frac{x}{\text{stride}} \times \text{stride} - x \qquad (4)$$

$$\text{Padding\_y\_yellow} = \frac{y}{\text{stride}} \times \text{stride} - y \qquad (5)$$

2. Fill 2 (blue part): Fill the outer border with 1/2 sliding step size (considering the expansion prediction of edge data). The filling effect is illustrated in Fig. 15.

$$\text{Padding}_{\text{blue}} = \frac{stride}{2} \qquad (6)$$

3. Only the prediction result of the center stride × stride of the Sliding window is reserved for each prediction, as shown in Fig. 16.

## RESULTS AND DISCUSSIONS

### Contribution to semantic segmentation of remote sensing

Data preprocessing is essential before model training. A superior processing method effectively improves model accuracy. Because the sensor records from orbit, the imaging process will inevitably be affected by many factors, such as the action of external radiation and the atmosphere, resulting in a degree of error between the remote sensing image and the original view. Here, we adopt radiometric calibration and atmospheric correction to eliminate maximally the influence of error. There are usually many different bands in remote sensing images, and different bands are combined depending on the land parcel category. The practice of band combination in this study will provide a reference for the field of remote sensing parcel segmentation. Panchromatic band images have the advantage of higher resolution. Further to enhance the information contained in the

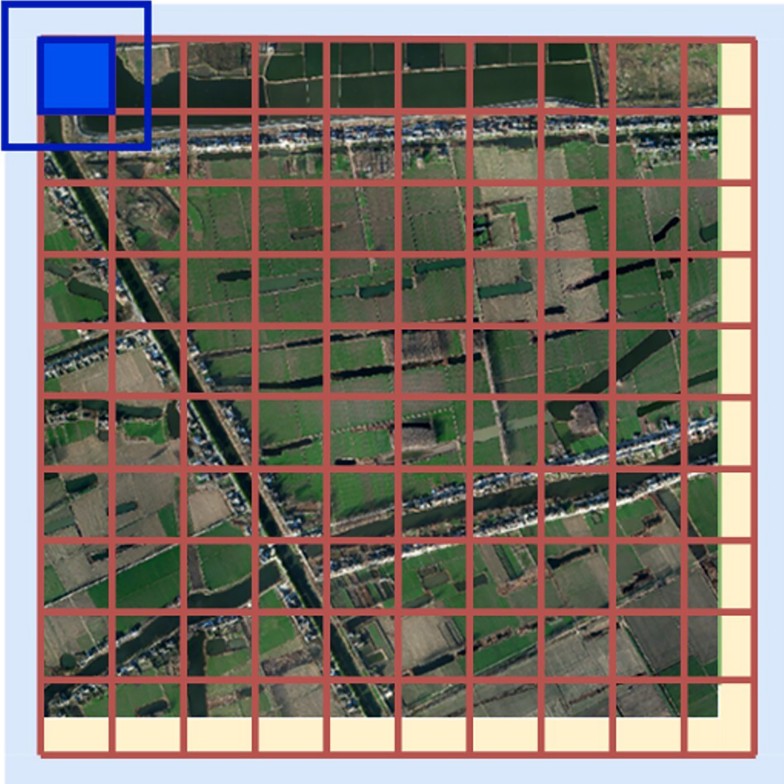

**Figure 15 Filling diagram.** 

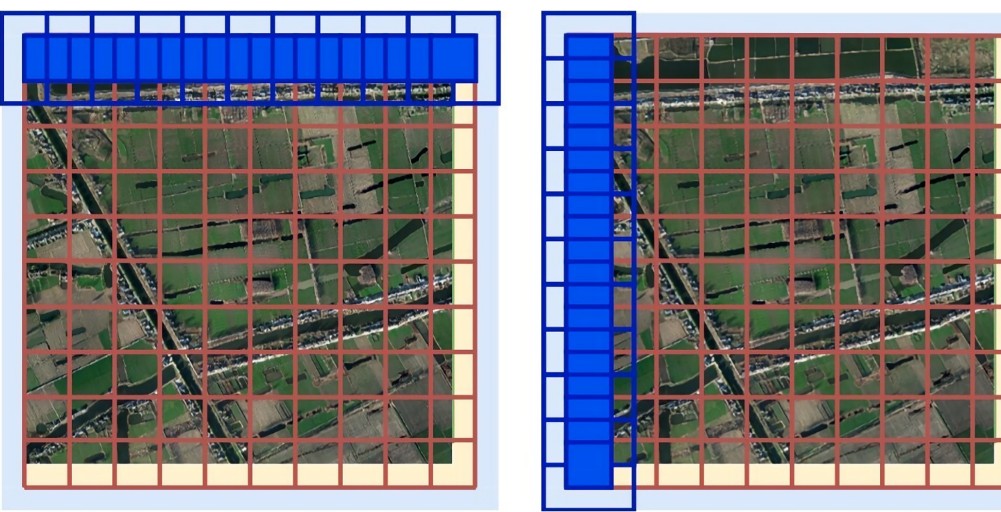

**Figure 16 A sketch of a sliding window.** The horizontal sliding window predicting process is on the left and the vertical sliding window predicting process is on the right.

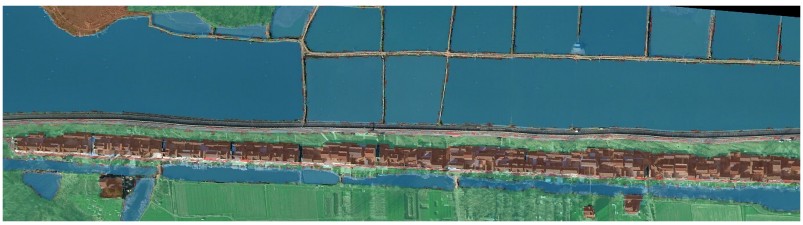

**Figure 17** OCRNet prediction map.

image, we fuse the selected multispectral band data with panchromatic band data. In this study, blocks of vegetation, water, and roads are highlighted by the Brovey transform fusion method to improve segmentation accuracy. This paper provides a reference for the selection of image fusion data sources and fusion methods.

The dataset used in this experiment includes five categories: forest, building, water body, rationale, and other categories. After training with GCNet, PointRend, OCRNet, DANet, APCNet, and other semantic segmentation networks, it was found that the pixel points of the roads category were mainly distributed in thin strips and had fewer numbers than other categories when inferring the prediction images. The OCRNet model prediction map for a remote sensing image is shown in Fig. 17.

The main reason for the above results is that in the process of feature extraction for images, roads are mainly strip-shaped and occupy a relatively small area. If a larger general symmetric convolution kernel is used to extract features from images, the effect may change because the convolution kernel size is too large to achieve accurate feature extraction and other unnecessary information will be introduced. For recognizing and segmenting traditional remote sensing images the method of introducing an attention mechanism effectively improves the model training accuracy and can also solve a series of problems in object classification. The introduction of the attention mechanism could also alleviate problems in remote sensing image extraction. However, because the road shape is a strip, the image can be extracted using the asymmetric convolution core of the bar shape, and the attention mechanism module of CBAM can be used to obtain the target area requiring attention. Simultaneously, the two introduced structures can enhance the outcome of the entire training process. Therefore, the original AC-CBAM module was created. A comparison of the predictions is shown in Fig. 18.

The experiments show that the entire remote sensing image predicts well. This method effectively extracts a complete image. Overall, this study undertook the following research.

1. Introduction of related theory and technology: Related theories such as neural networks and their training processes were introduced to commonly used methods of remote sensing image segmentation.

2. Study area selection and remote sensing image preprocessing: To improve the image quality, remote sensing image is preprocessed.

3. Dataset preparation: Label data were generated by automatic labeling and manual validation for data preprocessed from remote sensing images, and the dataset is expanded to 10,000 pieces.

4. In the decoder part, we created the AC-CBAM module and added the h-swish activation function. The experimental results show that the performance of the model was somewhat improved.

5. Rethinking the network structure from object features: According to the experimental results, the AC-CBAM module and sliding window prediction can improve the accuracy of extraction to some extent, which proves that the combination of computer vision and remote sensing knowledge will bring new ideas to solve many problems in the field of remote sensing.

## Different from existing methods

The CBAM has proven to be an attention mechanism module that can be widely used in target detection and semantic segmentation tasks. However, through experiments, we found that the unmodified CBAM is not suitable for the semantic segmentation of remote sensing images. Because the CBAM is an attention mechanism module based on ordinary convolution, a symmetrical square convolution kernel was used in feature extraction, whereas the improved AC-CBAM used a strip asymmetric convolution kernel, which is more in line with the category characteristics in the experimental task. When extracting features from the convolution kernel, most of the information obtained in the convolution area is acquired from categories; thus, when integrating information, the variance in different information is greater and more conducive to identification.

We obtained the results of semantic segmentation of OcrNet, Deeplabv3+, and other networks, and compared the effects of each method. First, we used OcrNet. The purpose of the object context information proposed by OcrNet is explicitly to enhance the object information. First, the regional feature expression of a group of objects is calculated, and then the representation of these regional-feature objects is propagated to each pixel according to the similarity between the regional-feature object's representation and the pixel feature's representation, to achieve the intended effect. However, for the category of very narrow roads, the advantages of OcrNet were not fully reflected in this experiment. Then, we used Deeplabv3+, which introduces hole convolution, for each convolution output to contain a large range of information without losing any. However, a large range of information does not necessarily contain sufficient information, and will not meet the experimental requirements. Thereafter, we continued to use other methods, and finally, through experimentation, obtained the best effect with the DNLNET method and further improved it.

## Limitations and future work

The image quality extracted from satellites is limited by the complexity of cloud and fog masking geographical features which reduces the accuracy of the obtained images. Owing to the irregular distribution of plots and the mutual embedding of various plots, the traditional convolutional neural network cannot better extract the edge feature

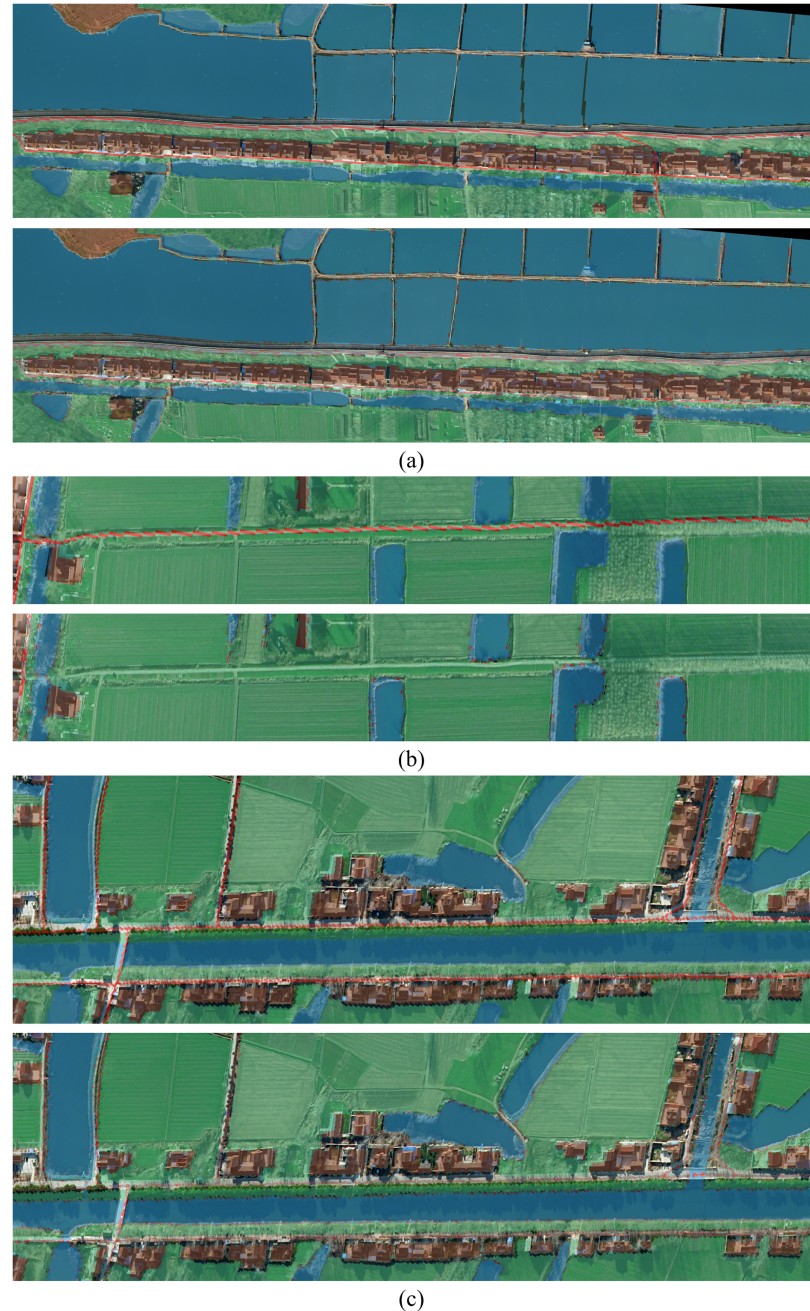

**Figure 18 The comparison groups of four different regions.** (A–C) The comparison groups of three different regions. For each group, the figure above is the ACB prediction chart and the figure below is OCRNet prediction chart.

information of junctions, thus having a considerable impact on the model's recognition capability.

In the future, this research will further improve the channel separation attention-detection network. There are two approaches to improving the network structure: one is

the enhancement algorithm of defogging by fusing the improved dark channel and Retinex (*Zhou, Chen & Sun, 2014*). By using the single-scale Retinex algorithm to enhance the restored image the problem of low accuracy caused by the incorrect estimation of transmittance could be somewhat alleviated. The enhanced image is more consistent with human visual characteristics, and the image details are more obvious. Another direction is to introduce a deformable convolution (*Dai et al., 2017*). In this method, the fixed-shape convolution matrix is transformed into a variable convolution matrix that can adapt to the shape of the object for the network structure itself to adapt. In addition, the edge information of the image is better extracted further to improve the fitting effect of the model.

### Funding
The authors received no funding for this work.

### Competing Interests
The authors declare that they have no competing interests.

### Author Contributions
- Zhao Shun conceived and designed the experiments, performed the experiments, analyzed the data, performed the computation work, authored or reviewed drafts of the paper, and approved the final draft.
- Danyang Li performed the experiments, performed the computation work, authored or reviewed drafts of the paper, and approved the final draft.
- Hongbo Jiang conceived and designed the experiments, performed the experiments, performed the computation work, authored or reviewed drafts of the paper, and approved the final draft.
- Jiao Li performed the computation work, prepared figures and/or tables, authored or reviewed drafts of the paper, and approved the final draft.
- Ran Peng conceived and designed the experiments, performed the experiments, analyzed the data, performed the computation work, authored or reviewed drafts of the paper, and approved the final draft.
- Bin Lin conceived and designed the experiments, performed the experiments, analyzed the data, performed the computation work, prepared figures and/or tables, authored or reviewed drafts of the paper, and approved the final draft.
- QinLi Liu analyzed the data, prepared figures and/or tables, and approved the final draft.
- Xinyao Gong analyzed the data, prepared figures and/or tables, authored or reviewed drafts of the paper, and approved the final draft.
- Xingze Zheng performed the experiments, analyzed the data, prepared figures and/or tables, and approved the final draft.
- Tao Liu performed the computation work, authored or reviewed drafts of the paper, and approved the final draft.

## Data Availability

The data is available at figshare: Bin, Lin (2021): 512.rar. figshare. Figure. https://doi.org/10.6084/m9.figshare.16767214.v1.

The code is available at GitHub: https://github.com/LinB203/remotesense.

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
