# Peer review of "Research on remote sensing image extraction based on deep learning"

_PeerJ Computer Science, doi:10.7717/peerj-cs.847_

## Round 0.1 · original submission · Major Revisions

Based on the comments of the reviewers, I would like to invite you to revise your paper. The writing should be improved. Please address the reviewers' concerns and prepare a detailed response letter. I do not expect you to cite any references recommended by reviewers unless you feel they are relevant.

Reviewer 1 ·

Basic reporting

This paper mainly focused on the classification of remote sensing images by carrying out atmospheric calibration, band combination, image fusion and other data enhancement methods for Landsat 8 satellite remote sensing data to improve the data quality. In addition, deep learning and spatial/channel attention are applied to remote sensing image block segmentation in this paper. Although the experiments show the effectiveness of this method, there are still some major concerns that need to be carefully clarified and revised before considering a possible publication.

Experimental design

1.The proposed method consists of several important components, so please add ablation experiments to evaluate the importance of each component. How each component contributes to the final performance gain? Especially the AC-CBAM compared to CBAM.
2.“Results and discussions” section is too short. Please give more experimental results and deep discussions of the results. Especially the deep reasons for the gains of the results.

Validity of the findings

1.There exist many grammar errors throughout the paper, which severely disturbs the readability. I strongly suggest the author to carefully revise the paper for English grammar, the choices of words, the sentence structure, and the use of articles (a native speaker is strongly recommended for this task).
2.In my opinion, the novelty of this paper is somewhat marginal, so the authors need to describe the main contributions of this paper clearly in the Introduction section.
3.What is the full name of AC-CBAM? Abbreviations should be defined at first mention and used consistently thereafter.
4.The authors should clearly highlight the difference between CBAM and AC-CBAM.
5.A deep literature review should further given, especially regarding the topics involved in this paper. Therefore, the reviewer suggests discussing the advances by citing some references, e.g., “When deep learning meets metric learning: remote sensing image scene classification via learning discriminative CNNs”, “Remote sensing image scene classification meets deep learning: challenges, methods, benchmarks, and opportunities”, and “Feature enhancement network for object detection in optical remote sensing images”.
6.There is a typo in the caption of Table 6.

Annotated reviews are not available for download in order to protect the identity of reviewers who chose to remain anonymous.

Reviewer 2 ·

Basic reporting

The traditional remote sensing image segmentation technology cannot make full use of the rich spatial information of the image, the workload is too large, and the accuracy is not high enough. Aiming at these problems, this paper carried out atmospheric calibration, band combination, image fusion and other data enhancement methods for Landsat 8 satellite remote sensing data to improve the data quality. In addition, deep learning is applied to remote sensing image block segmentation in this paper. Based on the full convolutional neural network of codec structure, AC-CBAM innovative structure is proposed, and the optimization module of integrated attention and sliding window prediction method are adopted to effectively improve the segmentation accuracy. This paper has done some work, however, there are at least the following problems.

1) The format of English is nonstandard, for example, in Introduction, the first paragraph is too long, while other paragraphs are too short. In addition, in row 33,“In the experiment of test data, the models mIoU, mAcc and aAcc in this paper reach 97.34%, 98.66% and 98.67% respectively, which is 1.44% higher than DNLNet (95.9%), which provides a reference for deep learning to realize the automation of remote sensing land information extraction”, it is unusual for“ which” appearing twice continuously, etc.
2) The English expression is not accurate enough, much of the translations fails to conform to English expression and culture connotation, resulting in poor readability, and it is best to find professionals to improve the language expression. For example, in Row 50 of 4.3 Slide Window Prediction , “And many other constraints and prior information to improve the performance of semantic segmentation”. In row 290/ 291, “finally got good spatial resolution and retained the images of the multispectral image information, to ultimately improve the effect of plot extraction have good role in promoting” , etc.

Experimental design

no comment

Validity of the findings

no comment

Additional comments

no comment

---

## Round 0.2 · accepted · Accept

The revised paper has addressed the comments of previous reviewers. I recommend it for publication.